# Octadecanoids as emerging lipid mediators in cnidarian-dinoflagellate symbiosis
Marina T. Botana [1,2], Robert E. Lewis[1], Alessandro Quaranta [2], Olivier Salamin[2],
Johanna Revol-Cavalier[2,3], Clint A. Oakley [1], Ivo Feussner [4], Mats Hamberg[2,3], Arthur R. Grossman [5],
David J. Suggett [6], Virginia M. Weis[7], Craig E. Wheelock [2,8] ✉ & Simon K. Davy [1] ✉

Oxylipin signaling has been suggested as a potential mechanism for the inter-partner recognition and homeostasis regulation of cnidarian-dinoflagellate symbiosis, which maintains the ecological viability of coral reefs. Here we assessed the effects of symbiosis and symbiont identity on a model cnidarian, the sea anemone *Exaiptasia diaphana*, using mass spectrometry to quantify octadecanoid oxylipins (*i.e.*, 18-carbon-derived oxygenated fatty acids). A total of 84 octadecanoids were reported, and distinct stereospecificity was observed for the synthesis of *R*- and *S*-enantiomers for symbiont-free anemones and free-living cultured dinoflagellate symbionts, respectively. Symbiont-derived 13(*S*)-hydroxy-octadecatetraenoic acid (13(*S*)-HOTE) linked to a putative 13(*S*)-lipoxygenase was translocated to the host anemone with a 32-fold increase, suggesting it as a biomarker of symbiosis and as a potential agonist of host receptors that regulate inflammatory transcription. Only symbiosis with the native symbiont *Breviolum minutum* decreased the abundance of pro-inflammatory 9(*R*)-hydroxy-octadecadienoic acid (9(*R*)-HODE) in the host. In contrast, symbiosis with the non-native symbiont *Durusdinium trenchii* was marked by higher abundance of autoxidation-derived octadecanoids, corroborating previous evidence for cellular stress in this association. The putative octadecanoid signaling pathways reported here suggest foundational knowledge gaps that can support the bioengineering and selective breeding of more optimal host-symbiont pairings to enhance resilience and survival of coral reefs.

Ecological success of coral reefs stems from the symbiotic relationship between the cnidarian host (e.g., corals, sea anemones) and its dinoflagellate algal endosymbionts of the family Symbiodiniaceae. Evolution of this symbiosis over space and time has resulted in a high diversity of host and symbiont genotypes, as well as a high specificity of host-symbiont pairings and therefore, the occupation of distinct ecological niches by the holobiont (i.e., the whole symbiosis)[1–3]. The symbiotic dinoflagellates supply their host with products from photosynthesis, including sugars, lipids, and amino acids[4–6] and receive inorganic substrates and shelter from the host[7,8]. As in

other symbiotic systems (e.g., rhizobia-leguminous plants, human gut-microbiota), host-symbiont exchange not only involves nutritional compounds, but also cell signaling molecules that regulate inter-partner recognition and symbiosis function[9,10]. These signaling molecules might cross the symbiosome membrane, the host-derived vacuole membrane that encloses the dinoflagellate symbionts within the cnidarian host's gastrodermal cells, and which acts as the primary host-symbiont interface[11]. Glycans, small peptides, inositols, polar lipids, oxylipins and non-coding RNA have all been proposed as candidate signaling molecules that may

[1]School of Biological Sciences, Victoria University of Wellington, Wellington, New Zealand. [2]Unit of Integrative Metabolomics, Institute of Environmental Medicine, Karolinska Institutet, Stockholm, Sweden. [3]Larodan Research Laboratory, Karolinska Institutet, Stockholm, Sweden. [4]Department of Plant Biochemistry, Albrecht-von-Haller-Institute for Plant Sciences and Goettingen Center for Molecular Biosciences (GZMB), University of Goettingen, Goettingen, Germany. [5]Carnegie Institution for Science, Department of Plant Biology, Stanford, CA, USA. [6]KAUST Coral Restoration Initiative (KCRI) and Division of Biological and Environmental Science and Engineering (BESE), King Abdullah University of Science and Technology, Thuwal, Saudi Arabia. [7]Department of Integrative Biology, Oregon State University, Corvallis, OR, USA. [8]Department of Respiratory Medicine and Allergy, Karolinska University Hospital, Stockholm, Sweden.
✉e-mail: craig.wheelock@ki.se; simon.davy@vuw.ac.nz

interact with specific transmembrane transporters, enzyme-coupled receptors, and voltage-gated ion channels (e.g., G-protein coupled receptors, lectins, peroxisome proliferator-activated receptors) for mediation of symbiosis establishment and cellular homeostasis; however, the underlying regulatory mechanisms involved in these processes remain poorly understood[12].

Oxylipins are a group of metabolites derived from oxygenated fatty acids that have gained recent attention in the cnidarian-dinoflagellate symbiosis[12,13], while also being reported as important bioactive mediators of cellular function and fate in many biological systems[14]. Oxylipins can be synthesized by the oxidation of mono- and polyunsaturated fatty acids (MUFA, PUFA) either following release from the cell membrane by lipase hydrolysis (e.g., phospholipase A$_2$, PLA$_2$)[15] or by direct oxidation of the membrane lipids. They can also be formed by autoxidation processes (e.g., via free radicals), which are not stereoselective[16] (Fig. 1a). Enzymatic formation of oxylipins is generally selective for carbon and double bond positions within the precursor fatty acid, so that only a specific regioisomer (e.g., 9- or 13-hydroxy-octadecadienoic acid (HODE)) is synthesized by a given enzyme[17]. Biosynthesis of oxylipins is also often stereoselective[18], meaning that the geometrical configuration of the oxylipin product has a specific spatial 3D structure—for instance, either right-handed (the (R) enantiomer) or left-handed (the left (S) enantiomer) for molecules with only one chiral center[17] (Fig. 1b). As such, determining the chirality of an oxylipin can indicate the synthetic source of the compound (i.e., enzymatic or autoxidation). Determination of oxylipin biological function and regulation is aided by knowledge of the regio- and stereo-isomer configuration because the propagation of signaling cascades often relies on their interaction with stereoselective membrane receptors[19,20] (Fig. 1c).

To date, only oxylipins derived from oxidation of 20-carbon fatty acids (i.e., eicosanoids), particularly products of arachidonic acid (ARA, 20:4, n-6), have been suggested to exert key functions for host-symbiont communication in the cnidarian-dinoflagellate symbiosis[21–23]. Nevertheless, the high diversity of PUFA precursors, from both the host and symbiont, suggests that this symbiosis is a promising model for exploration of novel oxylipins and the pathways responsible for their synthesis[13,24,25]. PUFA oxidation is mediated by the availability of both the substrate (i.e., fatty acid)

and initiator (i.e., an oxygenating agent—enzymes, free radicals, or singlet oxygen), and by the scavenging capacity of the antioxidant machinery present in the biological system being studied. It is therefore necessary to explore the distribution of intact PUFAs and their modifying enzymes to better predict oxylipin profiles and their synthetic routes[26,27]. Stearidonic acid (SDA; 18:4, n-3) and octadecapentaenoic acid (ODPA; 18:5, n-3) are the most abundant PUFAs integral to the chloroplast membranes of the symbionts[13,28]. Oxylipins derived from 18-carbon fatty acids are termed octadecanoids[29] and have been primarily studied in plants, with the classical octadecanoid pathway critical for the formation of the phytohormone jasmonic acid (derived from alpha-linolenic acid, ALA 18:3, n-3)[30–32]. These ALA-derived metabolites are compounds of interest in the cnidarian-dinoflagellate symbiosis[33,34]. However, octadecanoids originating from SDA and ODPA are less studied and have only been reported to occur in cultures of Symbiodiniaceae[13]; their role in the cnidarian-dinoflagellate symbiosis remains unexplored.

Here, we assessed the effects of symbiotic state and symbiont identity in the cnidarian-dinoflagellate symbiosis using a chiral supercritical fluid chromatography (SFC) coupled to tandem mass spectrometry (MS/MS) method to perform metabolic profiling of octadecanoids[35]. We used the sea anemone *Exaiptasia diaphana* (commonly called "Aiptasia")—a widely used model system in cnidarian-dinoflagellate symbiosis studies[36]—when in symbiosis with either its native (homologous) symbiont *Breviolum minutum* or the non-native (heterologous) symbiont *Durusdinium trenchii*. This latter symbiont was chosen because it is a widely studied opportunist that provides less nutritional benefit to the host and induces cellular stress when it populates Aiptasia under experimental conditions[22,37]. Symbiont-free anemones (i.e., aposymbiotic) and cultured symbionts were also analyzed to determine species specificity for the biosynthesis of specific oxylipin stereoisomers before addressing the impact of symbiotic state. Our comprehensive phenotypic analyses combined with top-down bioinformatics for determination of candidate biosynthetic enzymes of 18-carbon fatty acids enabled us to further describe the signaling cascades and pathways present in the cnidarian-dinoflagellate symbiosis and provide evidence for a potential inter-partner exchange of octadecanoids and their potential roles as lipid mediators.

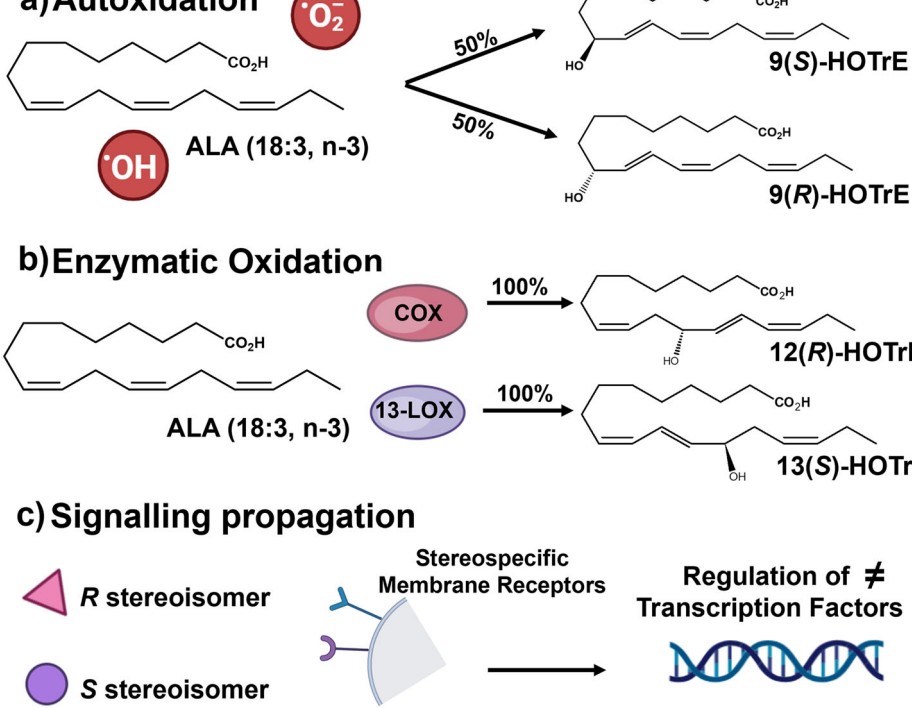

**Fig. 1 | Octadecanoid oxylipins can be formed *via* multiple synthetic routes. a** Autoxidation in which free radicals (*e.g.*, hydroxyl radical (•OH and superoxide (•O$_2^-$)) or singlet oxygen oxidize the fatty acid substrate. Notably, this process results in a racemic mixture of the resulting mono-hydroxyl species. Here we only represent the products for the 9-regioisomer, but the same process occurs for the 9-, 10-, 12- and 15- regioisomers (**b**) Enzymatic formation can result in stereo-selective formation of mono-hydroxy species (i.e., *R* vs *S* enantiomer). **c** Propagation of signaling cascades can rely on binding with stereoisomer-specific receptors. ALA alpha-linolenic acid; HOTrE hydroxy-octadecatrienoic acid, COX cyclooxygenase, 13-LOX 13-lipoxygenase.

## Results

### Distinct biosynthesis patterns of oxylipins in aposymbiotic anemones vs. cultured dinoflagellates

By chiral supercritical fluid chromatography (SFC) coupled to tandem mass spectrometry (MS/MS) (SFC-MS/MS), we were able to quantify 84 octadecanoids across all host and symbiont samples, including compounds derived from oleic acid (OA) (18:1, n-9), linoleic acid (LA) (18:2, n-6), alpha-linolenic acid (ALA) (18:3, n-3), gamma-linolenic acid (GLA) (18:3, n-6) and stearidonic acid (SDA) (18:4, n-3). There was significant structural diversity in the detected octadecanoids (see Fig. 2 for octadecanoid nomenclature); however, the overall concentrations of the different functional groups were in the order of mono-hydroxy>epoxy>oxo>trihydroxy>dihydroxy (Fig. 3). Diols derived from OA and ALA, as well as triols from ALA were absent from all sample groups (a detailed list of octadecanoids and their concentrations are provided in Supplementary Data 1).

Aposymbiotic anemones and cultured dinoflagellates were first assessed for species-specificity with respect to the biosynthesis and potential origins of octadecanoids before evaluating the effects of symbiotic state. Differential stereochemistry specificity was observed (Fig. 3a), with aposymbiotic anemones predominantly containing the (R) enantiomer of mono-hydroxy octadecanoids and cultured dinoflagellates the (S) enantiomer (Fig. 3b, c). Aposymbiotic anemones primarily produced (R) monohydroxylated forms of all regioisomers of SDA (i.e., HOTEs), GLA (i.e., HOTrEs-γ), and ALA (i.e., HOTrEs). The observed stereoselectivity was determined using the % enantiomeric excess (ee), with values ranging from 64 to 100% (Supplementary Data 2). Cultured *B. minutum* and *D. trenchii* exclusively contained the SDA-derived monohydroxy 13(S)-HOTE (ee = 100%), consistent with enzymatic biosynthesis. In both dinoflagellate species, ee values also suggested the biosynthesis of the 13(S)-HOTrE enantiomer (ee = 100% and 94% in *Breviolum minutum* and *Durusdinium trenchii*, respectively) (Supplementary Data 2). The only octadecanoid that

exhibited the same stereospecificity for both aposymbiotic anemones and cultured dinoflagellates was 9(R),10(S),13(R)-TriHOME (ee = 100% in all cases for position C-9) (Supplementary Data 2).

### Alterations in octadecanoid pathways in response to symbiotic state

Symbiont population densities in symbiotic anemones were not significantly different between the two dinoflagellate species (0.60 ± 0.16 and 0.57 ± 0.12 symbiont cells/ng anemone protein for *B. minutum* and *D. trenchii*, respectively). Any differences in the octadecanoid profile were therefore likely to be symbiont species-specific rather than a symbiont density effect. Profiling oxylipins with regio- and stereo-specificity enabled us to investigate alterations in distinct oxygenation pathways associated with symbiotic state and symbiont identity. There was a greater shift in the octadecanoid profile associated with symbiotic state (i.e., symbiosis vs. aposymbiosis) than with symbiont identity (Fig. 4). Phenotypic remodeling was broadly manifested as upregulation of octadecanoids in both the host and symbiont when in symbiosis. Additionally, both partners showed lower ee values for multiple octadecanoid enantiomers when in symbiosis than in their isolated states (Supplementary Data 2), with S enantiomers upregulated in the host and R enantiomers upregulated in the symbiont. Specifically, in symbiosis, the host exhibited an increased abundance of the 9(R),10(S),13(R)-TriHOME diastereoisomer and the (S) stereoisomers of HOTEs and HOTrEs-γ, with the biggest increases observed in *D. trenchii*-colonized anemones (p < 0.05 with FDR) (Fig. 5a and Supplementary Data 3). For EpODEs, EpOMEs, and epoxy-HOMEs, an increase of up to 30-fold only occurred in anemones containing *D. trenchii* (Supplementary Data 4). In contrast, in *B. minutum*-colonized anemones, the level of 9(R)-HODE decreased 3-fold relative to aposymbiotic anemones (Fig. 5a). In the symbionts, the abundance of all octadecanoids except for 13(S)-HOTE was upregulated in symbiosis vs. culture, with most increases being more marked in *D. trenchii* than in *B. minutum* (Fig. 5b and Supplementary

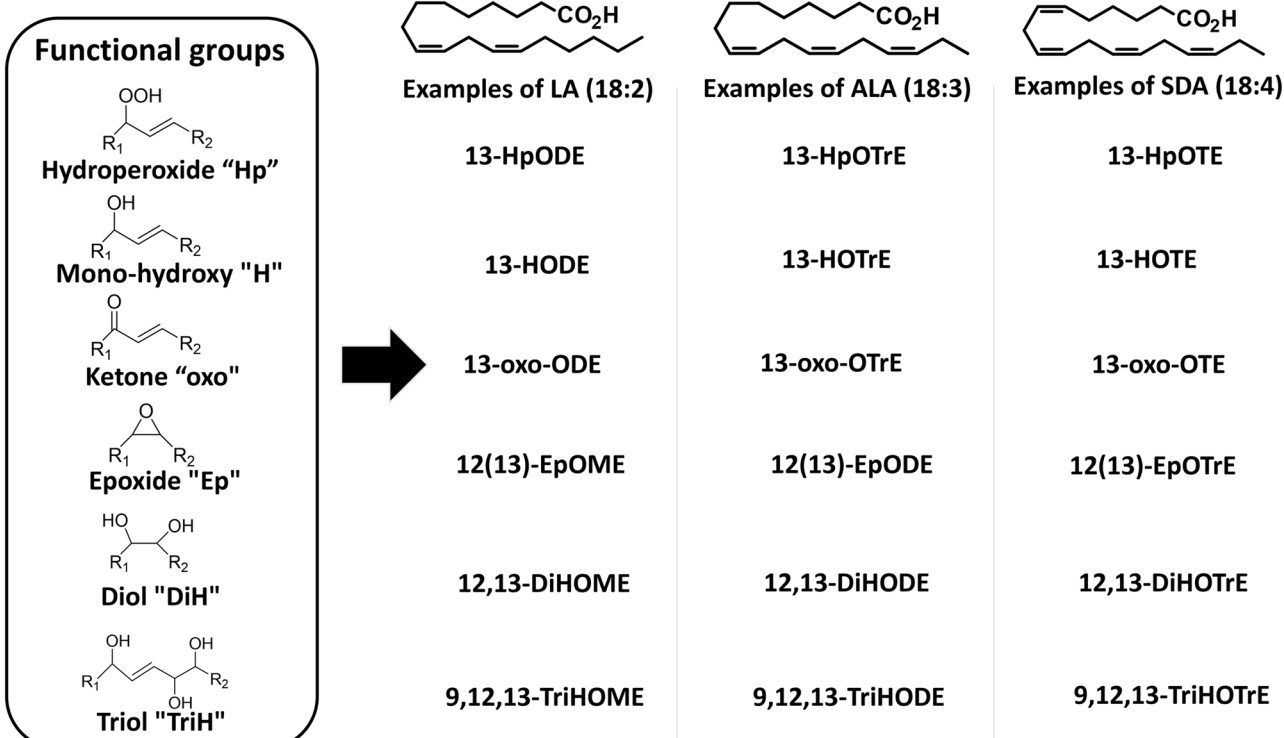

**Fig. 2 | Abbreviated nomenclature system for octadecanoids.** Compound names start with the positioning of the oxygenated moiety on the carbon backbone. The number(s) is followed by the abbreviation of the moiety ("Hp" for hydroperoxides, "H" for mono-hydroxy, "DiH" for diols, 'TriH' for triols, "oxo" for ketones, and "Ep" for epoxide) and by the alkyl chain length (for octadecanoids, a capital "O") and number of unsaturations ('M': mono-unsaturated; 'D': di-unsaturated; 'Tr': tri-unsaturated; 'T': tetra-unsaturated). The last letter indicates whether the compound is unsaturated ('E' for enoic, unsaturated; 'DA' for decanoic, saturated).

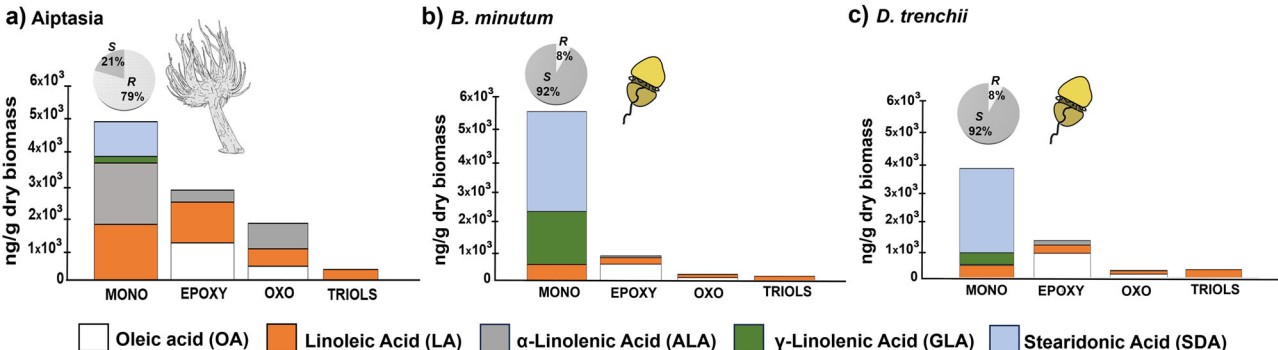

**Fig. 3 | Quantitative overview of all octadecanoids in Aiptasia and cultured symbionts.** Quantitative overview of all quantified octadecanoids in the: (**a**) Aposymbiotic Aiptasia host; and cultured (**b**) *Breviolum minutum* and (**c**) *Durusdinium trenchii* symbionts. Octadecanoids are grouped as: mono-hydroxy octadecanoids (MONO = 42 compounds), epoxides (EPOXY = 26 compounds), ketones (OXO = 9 compounds), and triols (TRIOLS = 5 compounds). Percentages of the abundance of the *R* and *S* enantiomers were calculated for mono-hydroxy octadecanoids to highlight the distinct stereochemistry specificity between aposymbiotic anemones and dinoflagellate symbionts. Octadecanoid levels are presented as ng of octadecanoid per weight of dry biomass. The abundance of diols (n = 2) was below 100 ng/g for all cases, and so data for these compounds are not presented in the graphs. Abbreviations: OA(oleic acid, 18;1, n-9), LA (linoleic acid, 18:2, n-6), ALA (alpha-linolenic acid, 18:3, n-3), GLA (gamma-linolenic acid, 18:3, n-6), SDA (stearidonic acid, 18:4, n-3).

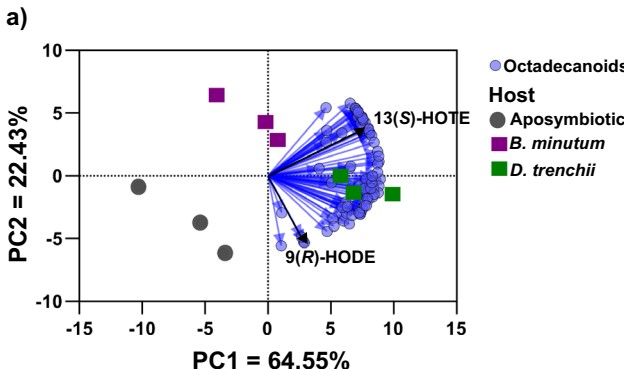

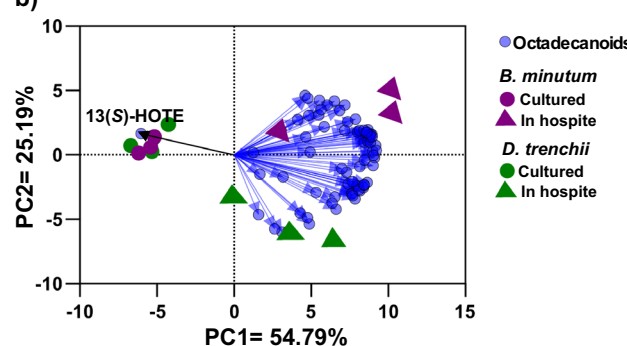

**Fig. 4 | Changes to octadecanoid profiles caused by symbiotic state.** Principal component analysis of changes to the octadecanoid profile of (**a**) Aiptasia host and (**b**) Dinoflagellate symbionts caused by symbiotic state. The sea anemone Aiptasia was either aposymbiotic or colonized by the native symbiont *Breviolum minutum* or non-native symbiont *Durusdinium trenchii*. Blue circles represent individual octadecanoids, and vector length is proportional to the individual contribution of each octadecanoid to the spatial distribution of sample replicates and groups. Compounds of interest are highlighted on the vectors and were further validated with a univariate test (see Table S3 and Fig. 5). 13(*S*)-HOTE = 13(*S*)-hydroxy-octadecatetraenoic acid; 9(*R*)-HODE = 9(*R*)-hydroxy-octadecadienoic acid.

Data 5). Increased enzymatic oxidation of ALA by 9-lipoxygenase (9-LOX) was evident for both dinoflagellate species. The abundance of HOTrEs from ALA was also elevated for both species. The abundance of epoxy-HOMEs and EpODEs increased in both species, but in *D. trenchii* this increase was 3-fold greater than in *B. minutum* (Fig. 5b and Supplementary Data 5).

**Evidence for potential inter-partner exchange of octadecanoids**
Aposymbiotic anemones did not contain 13(*S*)-HOTE, although this compound appeared in the host when colonized by either symbiont species, with the concentration 4-fold greater in anemones containing *D. trenchii* than those containing *B. minutum* (Fig. 5a; *p* < 0.05, with FDR). The (*S*) **ee** was 22% and 29% for anemones symbiotic with *B. minutum* and *D. trenchii*, respectively (Fig. 5). In comparison, the abundance of 13(*S*)-HOTE was significantly diminished in the symbiont when in symbiosis, in contrast to the pattern observed for all other octadecanoids (Fig. 5b and Supplementary Data 5). The (*S*) **ee** values decreased from 100% to 10% in *B. minutum* and from 100% to 23% in *D. trenchii* in the symbiotic state (Supplementary Data 2). These patterns suggest potential translocation from symbiont to host when in symbiosis.

**Novel 9(*S*)- and 13(*S*)-lipoxygenases in Symbiodiniaceae**
One *B. minutum* (MSTRG. 59407) and two *D. trenchii* (MSTRG.68745/ 7968) novel LOX-domain encoding mRNA sequences were proposed as novel LOX enzymes for Symbiodiniaceae (Fig. 6). The sequence for *B. minutum* was incomplete, but the alignment with AtLOX3 showed a combination of alanine and phenylalanine residues at the active site (Fig. S2), which revealed it as a 13(*S*)-LOX. MSTRG.59407 had no blast hits with identity percentages above 79.9% in any of the searched public databases (i.e., GeneBank, Reef Genomics and UNIPROT). The sequences for *D. trenchii* were 99.9% identical to two other sequences published in GenBank (CAK8999115.1 and CAK9070246.1), which have been described as "unnamed proteins". The alanine (Coffa determinant) and valine (Hornung site) alignments with the active site residues of AtLOX 1 and 5 revealed that MSTRG.68745/7968 were 9(*S*)-LOXes (Fig. S3). All transcripts demonstrated active constitutive expression across all tested samples of both cultured and symbiotic Symbiodiniaceae RNA-seq datasets (Fig. S4). Homology comparisons grouped the new Symbiodiniaceae LOXes within their own clade in the tree of life, being evolutionarily divergent from green algal (i.e., *Chlamydomonas reinhardtii* and *Lobosphera incisa*) and vascular plant LOXes, but rather closely related to fungal Mn-LOX sequences (Fig. 6).

**Discussion**
Interest in oxylipins as potential lipid mediators of inter-partner signaling in the cnidarian-dinoflagellate symbiosis has increased recently[12,13]. Here, octadecanoid biosynthesis by aposymbiotic anemones was found to be restricted to (*R*) enantiomers (as indicated by **ee** values of 100% for most

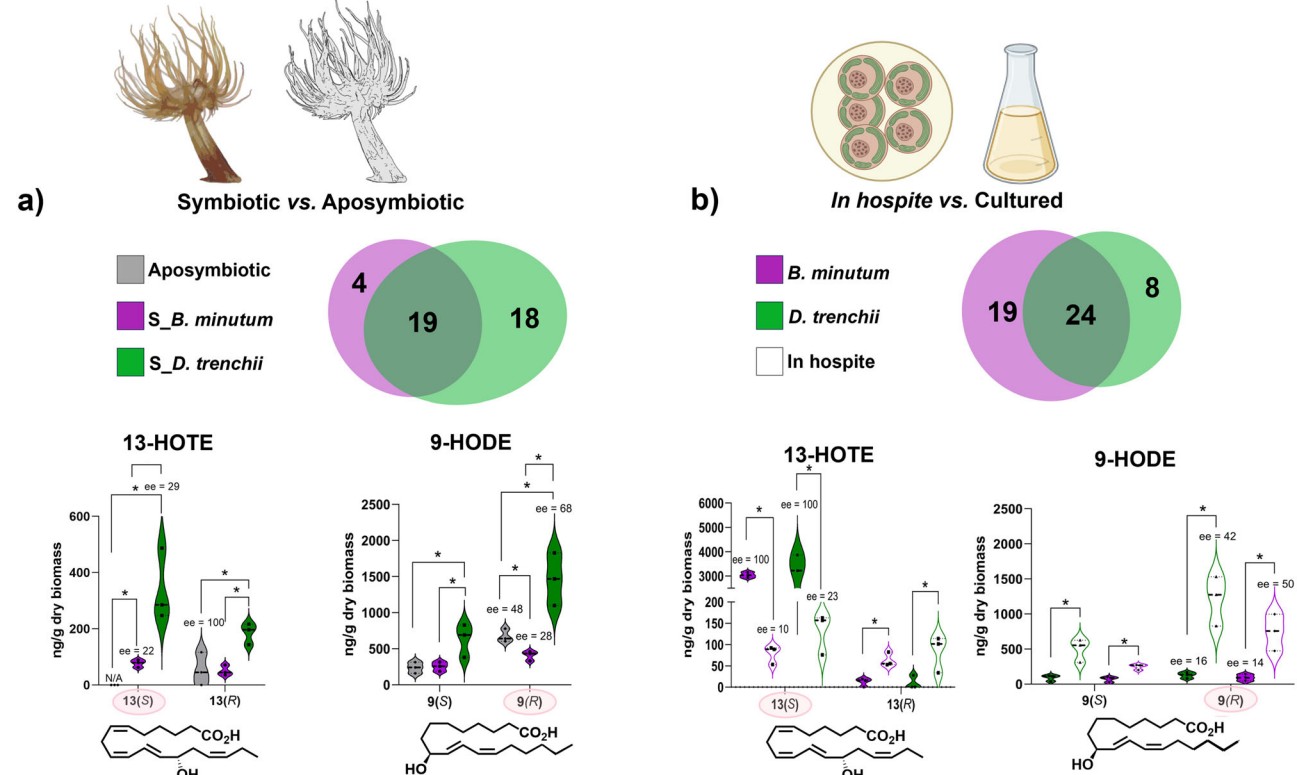

**Fig. 5 | Alterations to octadecanoid pathways in response to symbiosis. a** In the sea anemone Aiptasia when symbiotic with the dinoflagellate symbionts *Breviolum minutum* and *Durusdinium trenchii* (Symbiotic vs. Aposymbiotic); and (**b**) In the symbionts (*In hospite* vs. Cultured). Venn diagrams indicate the number of octadecanoids significantly altered when in symbiosis, including compounds shared by both species with pairwise fold-change differences > 2; *p* < 0.05 with FDR (see Tables S4 and S5 for details). Species-specific changes are color-coded. The enantiomeric excess values (**ee** %) for the *S* and *R* enantiomers of 13-HOTE and 9-HODE are shown in each graph. Box plot ranges represent the interquartile space in addition to the maximum and minimum whiskers with the median lane for each graph individually, explaining the differences in box plot total length among graphs. Symbiont cells in (**b**) were obtained from BioRender.com (Created in BioRender. Tb, M. (2025) https://BioRender.com/68xxjhx).

HOTEs, HOTrEs and HOTrEs-γ). This trend was also reported for eicosanoid biosynthesis in various soft corals[38–41] and the stony corals *Acropora* sp. and *Galaxea fascicularis*[24]; however, the stereospecificity of octadecanoid formation in cnidarians was previously unknown. In contrast, octadecanoid biosynthesis in both the cultured homologous symbiont *Breviolum minutum* and the heterologous *Durusdinium trenchii* was restricted to (*S*) enantiomers, with 13(*S*)-HOTE and 13(*S*)-HOTrE especially prominent (**ee** values from 94 to 100%). Both compounds are likely formed *via* a 13(*S*)-lipoxygenase (LOX), which is enantioselective for the synthesis of (*S*) octadecanoids in the jasmonic acid pathway[42]. In microalgae, studies of the LOX pathways have primarily focused on the synthesis of eicosanoids and docosanoids (derived from 22-carbon fatty acids)[43–45], and there is a paucity of investigations of octadecanoid biosynthesis[46,47]. To our knowledge, this is the first study of LOX enzymes in Symbiodiniaceae and applied top-down bioinformatics to obtain novel enzyme sequences putatively associated with octadecanoid biosynthesis in the cnidarian-dinoflagellate symbiosis.

Symbiosis resulted in increased levels of octadecanoids in both host and symbionts; however, there was a commensurate up to 3-fold decrease in their **ee** values. These findings suggest a potential dynamic inter-partner exchange of compounds and/or alterations to symbiosis homeostasis marked by a higher autoxidative state[48,49]. Specifically, *D. trenchii* promoted greater abundance of octadecanoids in the host, consistent with a stress response, which was further indicated by an up to 30-fold increase in epoxides (*i.e.*, EpODEs, EpOMEs and epoxy-HOMEs). In contrast, symbiosis with the homologous *B. minutum* resulted in only a slight increase in octadecanoid abundance, which was more consistent with a homeostatic adjustment. 9(*R*)-HODE has been linked to inflammation in mammals[50–52] and was exclusively downregulated in *B. minutum*-colonized anemones,

perhaps serving to dampen cellular stress and thereby contribute to the persistence of this host-symbiont pairing[2,22]. A putative regulatory role can be attributed to 9(*R*),10(*S*),13(*R*)-TriHOME (**ee** = 80% and 64% for position C-9, for anemones with *B. minutum* and *D. trenchii*, respectively), for which the abundance increased in symbiotic host tissues, although the associated pathway is unknown. In plants and humans, TriHOMEs have been linked to lipoxygenases with species and tissue stereospecificity[53–56]. In plants, Tri-HOMEs may also be linked to epoxyalcohol synthases (EAS), which together with allene oxide synthase (AOS), are from the CYP74 family (equivalent to cytochrome P450 family[42,57] in mammals and higher metazoans). In cnidarians, AOS and 8(*R*)- and 11(*R*) LOXes have been linked to eicosanoid synthesis[41,58–60]; but whether they are also involved in the synthesis of TriHOMEs and other octadecanoids remains to be investigated, as does the potential synthesis role by EAS.

Symbiosis also led to a general increase in the abundance of octadecanoids in the symbiont, with the greatest increase in *D. trenchii*. For both symbiont species, symbiosis caused changes in the ALA oxidation pathway, possibly *via* one of the linoleate-LOXes described here. This pathway occurs in plants[42], where synthesis of mono-hydroxy forms and a wide diversity of ALA downstream products (e.g., jasmonic acid and jasmonates) occurs in response to wounding and necrotrophic fungal infection and recognition[61], as well as during network communication in mycorrhiza[10,62]. The involvement of ALA oxidation products in host cell invasion and/or host-symbiont communication in the cnidarian-dinoflagellate symbiosis is therefore conceivable. However, the upregulation of LA and ALA oxidation pathways could also be linked to oxidative stress, with the increased abundance of EpOMEs and EpODEs in *D. trenchii* suggesting that this non-native symbiont species could be under greater cellular stress than the native

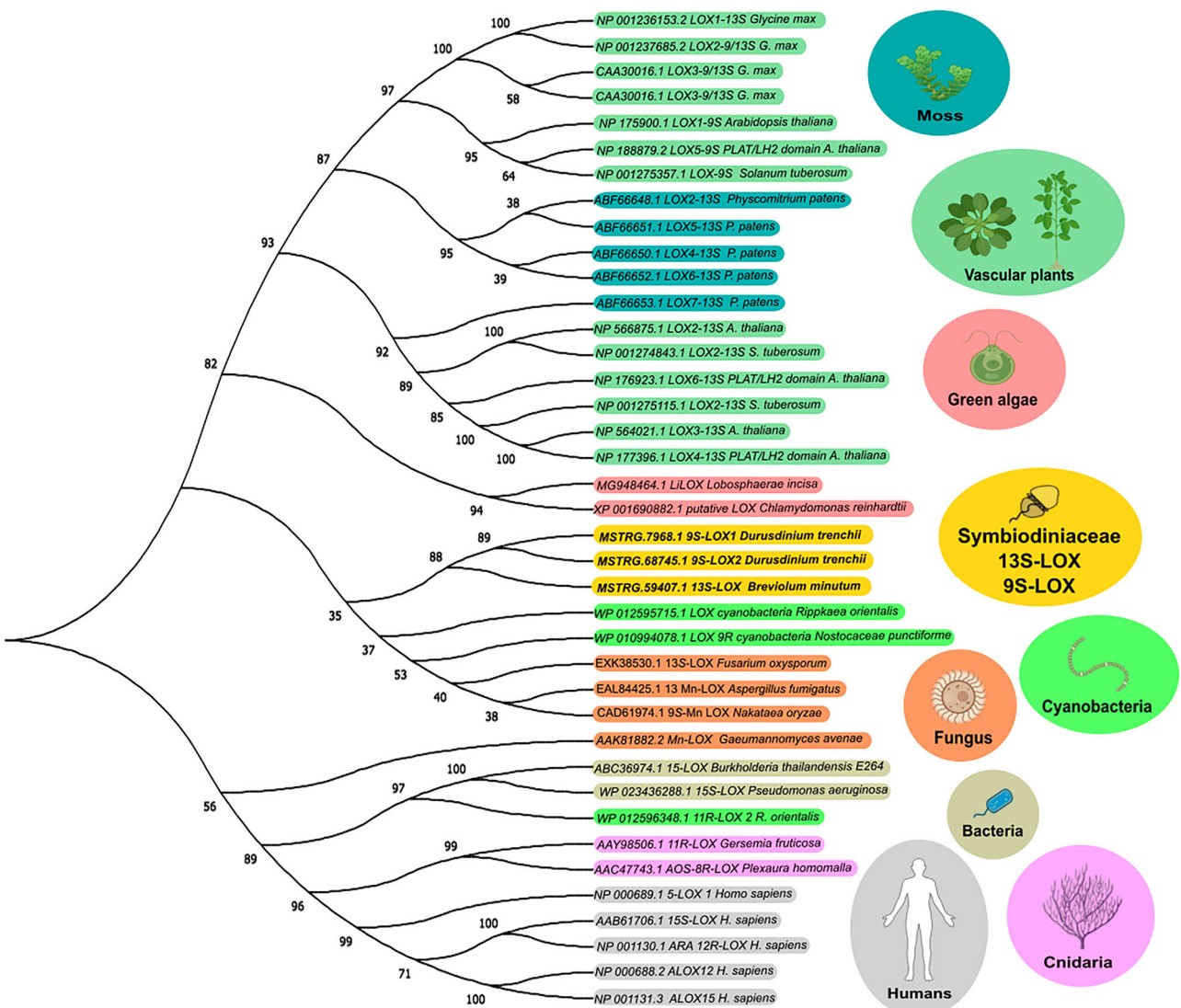

**Fig. 6 | Phylogenetic tree of lipoxygenases (LOXes) from different kingdoms and the novel LOXes in Symbiodiniaceae based on protein sequence homology.** Accession numbers were obtained from GenBank and are described in the Materials and Methods section. Figure was edited using Inkscape and BioRender (Created in BioRender. Tb, M. (2025) https://BioRender.com/68xxjhx).

*B. minutum*. Such stress could have arisen due to the greater levels of oxidative stress in the host's tissues discussed above, or perhaps other aspects of physiological dysfunction associated with the poorer degree of host-symbiont integration that is typical of this symbiotic pairing[2,22,63,64].

Regulatory signaling involves both intracellular and intercellular cascades, and the combination of both is crucial for endosymbioses[9,62,65,66]. In the cnidarian-dinoflagellate symbiosis, intercellular signaling between the partners may occur *via* G protein-coupled receptors associated with the symbiosome membrane[12,67], ultimately leading to alterations in host nuclear transcription factors[68,69]; however, the molecular pathways involved are poorly understood[48,70]. In the case of intracellular signaling, a given molecule may bind to nuclear peroxisome proliferator-activated receptors (PPAR) that also regulate transcriptional factors, such as nuclear factor kappa-B (NF-kB) and tumor necrosis factors (TNF), mediators of inflammatory and immune responses[66,71]. This latter mechanism has been widely studied in mammals but not in the cnidarian-dinoflagellate symbiosis. Nevertheless, the expression of both NF-kB and TNF has been observed to change in the host in response to symbiosis, leading to the suggestion that they help regulate symbiosis stability[48,70].

The role of octadecanoids as PPAR ligands and controllers of transcriptional factors has received considerable attention in the biomedical field, and they have been extensively targeted for the development of metabolic regulatory drugs[72]. 13(*S*)-HODE and 9(*S*)-HODE are PPAR-γ ligands and controllers of the immune response in mammals, showing that different oxylipins can bind to the same receptor, although binding affinity is variable[72,73]. The SDA-derived octadecanoid 13(*S*)-HOTE, extracted from cultures of the green microalga *Chlamydomonas debaryana*, has also been shown to be a strong PPAR-γ ligand and to promote downregulation of NF-kB and other transcription factors in cell cultures and in a murine recurrent colitis model[74–76]. In our experimental system, the abundance of the symbiont-derived 13(*S*)-HOTE decreased in the symbiont fraction and increased in the host fraction while in symbiosis, supporting evidence for inter-partner communication *via* a putative translocation of this compound. While confirmation is still needed, our findings support this octadecanoid-receptor binding as a potential mechanism of host immune modulation. Transport of oxylipins through membranes often occurs through lipoproteins and we have previously reported these proteins to be 20-fold more abundant when *Aiptasia* is colonized with *B. minutum*[77]. Ultimately, oxylipin-receptor binding and translocation are promising themes for future research on the cnidarian-dinoflagellate symbiosis, with considerable potential to draw parallels with the biomedical field given that the cnidarian immune system is similar to that of higher organisms, including mammals[78].

We also studied the enzymatic candidates for the biosynthesis of 13(*S*) octadecanoids, which included a novel 13(*S*)-LOX for *B. minutum* and two 9(*S*)-LOXes for *D. trenchii*. In vascular plants, most 9-LOXes described to date are cytosolic and only free fatty acids serve as substrate, whereas 13-LOXes are either chloroplast-derived or cytosolic; and both esterified membranes or free fatty acids can serve as substrates[42]. 13-LOX activity forms the 13-positional mono-hydroperoxy fatty acids (e.g., 13(*S*)-HpOTE and 13(*S*)-HpOTrE), whereas 9-LOXes synthesize the 9-positional forms (e.g., 9(*S*)-HpOTE and 9(*S*)-HpOTrE)[79]. These fatty acid hydroperoxides can serve either as substrates for additional metabolic transformation (e.g., AOS or EAS), or are reduced to form the corresponding mono-hydroxys (e.g., HOTrEs or HOTEs). In cyanobacteria LOX, both positional mono-hydroxy forms can be synthesized by the same enzyme, because the insertion of molecular oxygen into the fatty acid chain is guided by steric shielding of the radical intermediate rather than by the positional orientation of the fatty acid when entering the enzyme active site[80]. We do not know which mechanism occurs in the Symbiodiniaceae, but the novel LOX identified in *B. minutum* is a candidate for the synthesis of symbiont-derived 13(*S*) octadecanoids. The highly active constitutive expression of both 13(*S*)- and 9(*S*)-LOX candidates across both cultured and symbiotic Symbiodiniaceae RNA-seq datasets also suggests that they might not only be real sequences, but also of paramount importance for symbiont metabolism. Further characterization of the enzyme sequences presented here including exploration of the catalytic activity, regional specificity and fatty acid substrate preference are still needed for validation and might be promising for future research using a similar approach as in one of our previous studies characterizing a new LOX in *Lobosphaera incisa*[81].

In vascular plants, chloroplast-derived 13(*S*)-LOX has the highest substrate oxygenation kinetics with ALA[42,82], which is also true for LiLOX in the green alga *L. incisa*[81] and the cyanobacterium *Cyanothece* sp[80]. In plants, this initial oxygenation step is part of the jasmonic acid pathway, where jasmonic acid and other jasmonates help regulate the immune response, growth, and development[31,83]. In Symbiodiniaceae, SDA and ODPA are more abundant than ALA[13,28], suggesting that their as yet unidentified downstream products might be key metabolic mediators and perhaps play a role in host-symbiont recognition. This hypothesis needs further investigation; however, it suggests that future studies should focus on ODPA-derived octadecanoids as well as oxylipins from 20-carbon to 22-carbon PUFAs to provide a broader understanding of the lipid signaling pathways present in the cnidarian-dinoflagellate symbiosis. It would be valuable in future studies to also determine the PUFA content in both the symbiont and host, which could provide increased understanding of the biosynthesis pathways and potential inter-partner exchange and signaling of octadecanoids and other oxylipins.

This work presents the first detailed quantification of octadecanoids in cnidarians and their associated symbionts and reveals the stereospecificity of octadecanoid biosynthesis in the studied species. Determination of the octadecanoid stereochemistry provided insight into their biosynthetic route of formation. Our findings indicate a dynamic exchange of octadecanoids between symbiont and host and suggest that specific oxylipin stereoisomers act as putative lipid signaling mediators of symbiosis. Understanding the molecular processes that underpin host-symbiont communication and symbiosis stability is of importance across symbiotic systems and is key in determining how to stabilize or strengthen host-symbiont interactions. For this purpose, the combination of lipidomics and reverse genetics is a powerful tool for further elucidation and discovery of new molecular pathways, which can contribute to our improved understanding of coral reef function. Accordingly, increased efforts should be made in understanding octadecanoid formation, endogenous levels, and signaling in relation to host-symbiont communication. This approach has the potential to improve the development of restoration and conservation tools involving bioengineering and selective breeding of more optimal host-symbiont pairings to enhance coral reef survival in the face of environmental stressors.

# Material and methods

## Maintenance of animal and algal cultures

Symbiodiniaceae cultures (*Breviolum minutum*, culture ID CCMP830; *Durusdinium trenchii*, culture ID D1A001 (see Supplementary Data 6 for species confirmation) were grown for eight years in f/2-enriched 0.22 μm filtered seawater (FSW)[84]. Culture flasks were maintained inside an incubator set to 25 °C (±0.5 °C), under cool fluorescent lights (Osram Dulux 36/W890) at 90–110 μmol photons $m^{-2}$ $s^{-1}$ on a 12:12 h light/dark cycle.

Long-term clonal stocks of the sea anemone *Exaiptasia diaphana* (Aiptasia, culture ID: NZ1) were rendered aposymbiotic using a menthol protocol[85]. The anemones were confirmed to be aposymbiotic based on the absence of chlorophyll autofluorescence under confocal microscopy (Olympus Provis AX70; 100x magnification) and the absence of symbiont DNA amplification through polymerase chain reaction using ITS2_F and ITS_R primers[86]. Two thirds of the aposymbiotic stock (*n* = 150 anemones) were inoculated with either homologous *B. minutum* or heterologous *D. trenchii* and maintained in symbiosis for 15 months as described by Wuerz et al.[87]. All anemones were maintained in FSW and fed once a week with freshly hatched *Artemia* sp. nauplii. Stocks were kept at 25 °C and 90–110 μmol photons $m^{-2}$ $s^{-1}$ on a 12:12 h light/ dark cycle using Philips 6500 K bulbs.

## Experimental setup

Symbiodiniaceae cultures (*n* = 3 *per* species) were grown in 500 mL Erlenmeyer flasks under the same conditions for long-term maintenance. Initial cell densities in flasks were 1000 cells per mL and culture growth was monitored using the confocal imaging system IN Cell analyzer 6500 Hs, where cell density was quantified with Image J processing software (ten counts *per* replicate). Cultures were monitored over time, and sampling for octadecanoid analysis occurred after 16 days, while the cultures were still in exponential growth (Fig. S1), using sterile pipettes (150 mL *per* replicate). Samples were centrifuged at $4000 \times g$ for 5 min and the resulting algal pellets flash-frozen and freeze-dried before storage at −80 °C in sterile 1.5 mL Eppendorf tubes. On the day of sampling, cultures were dark acclimated for 15 min within 4 h of initiating the light cycle, before measurements of "photosynthetic health" ($F_v/F_m$) using a Diving Pulse Amplitude Modulated Fluorometer (Diving-PAM, Walz, Effeltrich, Germany; settings: measuring light = 4, saturation intensity = 8, saturation width = 0.8 s, gain = 3, and damping = 3) (Fig. S1).

Anemones for each experimental treatment (i.e., aposymbiotic; symbiotic with *B. minutum*; and symbiotic with *D. trenchii*) were split across $3 \times 400$ mL jars *per* treatment (*n* = 15 anemones *per* jar) that contained FSW. These jars were then transferred to water baths set at 25 °C (±0.5 °C) and under the same light conditions as described for the anemone stocks. All anemones were fed with *Artemia* sp. nauplii at the same time of day, and the old water was changed with fresh FSW 8 h after feeding[21,22]. Anemones were maintained under these conditions for 16 days prior to sampling and analyses. On the sampling day, $F_v/F_m$ of symbiotic anemones was measured (Fig. S1) as described above for the Symbiodiniaceae cultures. Anemones (*N* = 10 *per* jar) were then pooled, washed, and homogenized with 1 mL of ice-cold 10 mM sodium phosphate buffer (pH 7.4) containing 100 μM of the Fe chelator deferoxamine mesylate, using a glass tissue grinder. Aposymbiotic anemone homogenates were flash-frozen, freeze-dried, and stored at −80 °C in 1.5 mL sterile Eppendorf tubes. Homogenates from pooled symbiotic anemones were transferred to sterile Eppendorf tubes and centrifuged at $500 \times g$ for 5 min at 4 °C to separate host and symbiont fractions. Host tissue supernatant was transferred to a separate sterile tube, and the pelleted symbiont dinoflagellate fraction was washed free of residual host material by resuspension and centrifugation using the same buffer solution. A 20 μL aliquot of each host supernatant was analyzed for protein content using a fluorometric Qubit Protein Assay Kit. After resuspension for a second time, the symbiont fraction was aliquoted equally for cell counts, conducted using the confocal imaging system IN Cell analyzer 6500 Hs (as described above), and then normalized to host protein content (see Results).

All steps were performed on ice and final fractions were flash-frozen, freeze-dried and stored at −80 °C until further analysis.

## Sample preparation and extraction

Octadecanoids were extracted from freeze-dried biomass by adding 1.5 mL of methanol (MeOH) and 10 μL of internal standard (IS) mix (Supplementary Data 7 and 8). All samples were thoroughly mixed by vortexing and subsequently sonicated in an ice bath for 30 min. Following centrifugation for 10 min at $15,000 \times g$ at 4 °C, the supernatant was transferred to a glass tube and evaporated to dryness under nitrogen. To maximize compound recovery, this methanolic extraction was repeated a second time and combined with the first extract. The dried extracts were then reconstituted with 1 mL of a solution of 0.2 M $Na_2HPO_4$ and 0.1 M citric acid (pH 5.6). To concentrate the extract, solid phase extraction (SPE) was performed as described in Quaranta et al. (2022), using an Extrahera automated sample preparation system (Biotage, Uppsala, Sweden). Briefly, the 1 mL reconstituted extract was loaded onto a preconditioned 3 mL (3 cc/60 mg) Waters Oasis HLB cartridge (Milford, MA, USA). Samples were washed three times with 3 mL of HPLC-grade water and a fourth time with 3 mL MeOH:$H_2O$ (1:9). Octadecanoids were eluted with 2.5 mL of MeOH and the extract was further evaporated to dryness under a nitrogen stream. Reconstitution was performed with 80 μL MeOH, and the samples were then filtered through a 0.1 μm polyvinyl-i-dene fluoride membrane spin-filter (Amicon, Merck Millipore Cooperation, Billerica, MA, USA) before being transferred to liquid chromatography vials for analysis. Blanks comprised of FSW only, and culture medium was added to ensure that oxylipin phenotypic profiles in samples were not related to possible contamination from the experimental environment. All blanks were extracted in the same fashion as for the experimental samples.

## Octadecanoid profiling and analysis

A chiral supercritical fluid chromatography (SFC) coupled to tandem mass spectrometry (MS/MS) platform was used to perform quantitative metabolic profiling of octadecanoids as previously reported[35]. The published method was expanded by the addition of 27 novel custom synthesized standards derived from oleic acid (OA), alpha-linolenic acid (ALA, n-3), gamma-linolenic acid (GLA, n-6), and SDA (Supplementary Data 7). The discrimination of R and S enantiomers for the monohydroxy octadecanoids was based on the elution order in the chiral column, following the same patterns as observed for the 18 enantiopure standards detailed in Quaranta et al.[35].

Extracts were first analyzed using the SFC method on a Waters UPC2 system coupled to a Waters Xevo TQ-XS mass spectrometer. Chiral separation was performed on a polysaccharide Waters Trefoil AMY1 column (3.0 × 150 mm, 2.5 μm) set at 35 °C and an injection volume of 2 μL. In addition to supercritical $CO_2$ as the main eluent (phase A), a co-modifier consisting of MeOH:EtOH (8:2 by vol.) and $CH_3COOH$ 0.1% v/v was used as phase B. The gradient started with 5% B maintained until 1 min, before increasing it linearly to 25% B at 11 min and 30% B at 12.3 min. The column was then washed with 50% B for 2–5 min and re-equilibrated under the initial conditions for 2.2 min. Flow rate was 2.0 mL/min during separation and equilibration, but was decreased to 1.5 mL/min during washing. The active backpressure regulator (ABPR) was set to 2000 psi, and the make-up solvent consisting of MeOH and $CH_3COONH_4$ (5 mM) was flow rate-based on the co-modifier to avoid excess organic solvent at the source. It started at a flow rate of 0.2 mL/min before linearly decreasing to 0 mL/min after 6 min.

The chromatographic system had an MS source operating in negative-ion ESI mode with a capillary voltage of 1.9 kV, the source temperature at 150 °C, the desolvation temperature at 600 °C, the source offset at 30.0 V, the cone gas flow at 150 L h$^{-1}$, the desolvation gas flow at 1000 L h$^{-1}$, and the nebulizer gas pressure at 7.0 bar. Negative multiple reaction monitoring transitions along with the collision energy, cone voltage, and dwell time were manually optimized for each octadecanoid, and one transition *per* analyte was selected based upon sensitivity and selectivity. Calibration was

performed using an 11-point linear calibration model, applying a 1/x weighted least-squared regression. Retrieved octadecanoid quantity was normalized using the dry biomass, such that concentrations are reported in ng/g. MassLynx software version 4.2 was used for data acquisition and TargetLynx for data processing. Stereochemistry of the octadecanoids was evaluated using the enantiomeric excess (**ee**) (see IUPAC compendium for chemical terminology). For each characterized regioisomer with chiral configuration, the **ee** was calculated using the equation:

$$ee\% \text{ of } (R) \text{ or } (S) \text{ enantiomer} = (\%\text{major enantiomer}) - (\%\text{minor enantiomer})$$

## De novo transcriptome assembly and mRNA abundance quantification

To describe the transcriptomic responses of both cultured and symbiotic *B. minutum* and *D. trenchii*, a splice-aware transcriptome was assembled from previously sequenced RNA libraries (Maor-Landaw et al.[88], BioProject: PRJNA544863 and Bellantuono et al.[89], BioProject PRJNA508937). Using Fastp[90], the raw paired-end RNA sequence files were trimmed and filtered to remove sequencing adaptors and bases with Phred quality scores below 30. Reads with more than 30% low-quality bases, containing more than 5 N bases, and reads shorter than 50 bases after trimming were discarded.

Using the RNA-Seq aligner Spliced Transcripts Alignment to a Reference (STAR v2.7.10b[91]), all filtered sample reads were first mapped without guidance to the Aiptasia genome (assembly v1.1; GCF_001417965.1[92]) to filter anemone RNA reads and double mapping sequences for downstream expression quantitation. Un-mapped RNA reads were collected and mapped without guidance to the respective symbiont genomes of *B. minutum* (strain: Mf 1.05b.01; GenBank assembly: GCA_000507305.1[93]) and *D. trenchii* (strain: CCMP2556; GenBank assembly: GCA_963970005.1[94]). Splice-junction aware alignment outputs were assembled into sample transcriptomes using StringTie2 v2.0.3[95]. Sample transcriptomes were then merged into species RNA transcriptomes with a minimum transcript coverage of one mapped read *per* base pair. All assembled RNA transcript open reading frames (ORFs) greater than 100 amino acids were predicted using Transdecoder (TransDecoder.Predict v5.7, see Github in references), to predict the most likely protein coding sequences (CDs regions) and reduce the ORF false discovery rate. The ORFs were then filtered to retain only transcripts with either Blastp protein sequence homology (e-value < 1e$^{-5}$) to any described or hypothetical protein in a collection of Stramenopile and Alveolate genomes (Supplementary Data 9) and/or HMMER (hmmscan v3.4) Pfam domain homology. Internal unique transcripts were used as training data for the species-specific Markov Model-derived coding potential of the ORFs and to revise start codon choice where statistically appropriate. After prediction, the best ORF candidates with revised starts were propagated to each genome and written into new genomic protein-coding transcript annotations.

To quantify transcript abundance and test for differential expression, un-normalized read counts summarized at the gene (meta-feature) level for the homologous protein-coding transcript annotations were performed on the RNA sequence STAR genomic alignments for each sample, using FeatureCounts[96] (Rsubread v2.16.1). All counted genes were tested for differential expression among experimental conditions from the original raw datasets using DeSeq2[97], with default internal gene-filtering procedures (see Supplementary Data 10 forRNA-seq libraries used for LOX gene expression counts). Before expression testing, counted genes were pre-filtered to retain only genes with at least 10 un-normalized reads counted in every sample.

## Identification of Symbiodiniaceae LOX candidates and phylogeny

To identify lipoxygenase (LOX) sequences in the protein-coding transcriptomes, a collection of previously annotated LOX protein sequences, from both prokaryotic and eukaryotic species, was curated (Supplementary Data 11). All hypothetical Symbiodiniaceae protein sequences were scanned for sequence homology to the LOX references, using a Blastp cutoff of e-

value = $1e^{-4}$. Presence of Pfam LOX domains within the collection of Blastp candidates was confirmed with InterProScan (interproscan.sh v5.68-101.0)[98]. Protein sequence phylogeny of the 13($S$)-LOX candidates was constructed from MUSCLE multiple sequence alignments, together with the other LOX references[99]. The resulting distance matrix provided the information to generate a phylogenetic tree using the neighbor-joining clustering method, with branch lengths optimized *via* a phylogenetic maximum likelihood criterion[100], with Le and Gascuel model parameters[101]. The fitted tree was then bootstrapped 100 times.

## Statistics and reproducibility

Before evaluating the alterations caused by symbiotic state in both the host and symbiont, octadecanoid profiles of both aposymbiotic anemones and cultured dinoflagellates were characterized, and symbiont species-specific compounds and the potential origins of octadecanoid enantiomer biosynthesis patterns (i.e., *R* and/or *S*) were assessed by **ee** (Supplementary Data 2). Before statistical analysis, the octadecanoid dataset was normalized using log transformation. Principal component analysis (PCA) using Euclidian distance metrics was used to identify the main differences in the octadecanoid profiles associated with symbiotic state and symbiont identity, in the host and symbiont separately. A two-way crossed ANOVA using symbiotic state (Symb; two levels in the host: aposymbiotic and symbiotic; two levels in the symbiont: cultured and symbiotic) orthogonal to the dinoflagellate species (Symb; two levels: *B. minutum* and *D. trenchii*) was further applied for univariate validation of octadecanoids highlighted in the PCA. Additionally, pairwise comparisons were analyzed with Tukey's HSD test, and the significance threshold set at $p < 0.05$ with Benjamini-Hochberg false discovery rate corrections. A volcano plot analysis that considered pairwise fold-change differences of greater than two was also applied to the whole octadecanoid matrix, to investigate whether oxidation pathways were altered due to the symbiotic state in the host and symbiont. MetaboAnalyst version 6.0[102] was used for all statistical analyses. Octadecanoid quantity was normalized using dry biomass, such that concentrations are reported in ng/g. Boxplots in Fig. 5 represent the interquartile space in addition to the maximum and minimum whiskers with the median lane for each quantified octadecanoid and the individual data points are also presented ($n = 3$).

## Reporting summary

Further information on research design is available in the Nature Portfolio Reporting Summary linked to this article.

## Data availability

All relevant data generated and analyzed during this study can be found in Supplementary Information and in Supplementary Data 1–11.

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

## Acknowledgements

We thank Prof. Miguel Mies (University of Sao Paulo, Brazil), Dr. Matthias Kellerman (Carl-von-Ossietzky University, Germany), Dr. Matthew Nitschke, Dr. Gerhard Hagn and Prof. Robert Keyzers for reviewing the manuscript. We also thank Prof. Blair Paul (Marine Biological Laboratory, Woods Hole, USA) for the assistance provided with the bioinformatics pipeline and Dr. Ellen Hornung (Georg-August-University, Germany) for the assistance with the LOX analysis. This manuscript is in memory of Professor Raymond Valentine (University of California, Davis, USA), may his incessant curiosity in understanding life always remains with us. This research was supported by the Marsden Fund of the Royal Society Te Apārangi, grant number 19VUW086, awarded to S.K.D., C.A.O., A.R.G., D.J.S., and V.M.W., including a postgraduate scholarship awarded to M.T.B. I.F. acknowledges funding from the German Research Foundation (DFG) grant numbers GRK 1422 and GRK 2172. C.E.W. acknowledges support from the Swedish Research Council (2022-00796) and the Cayman Biochemical Research Institute (CABRI).

## Author contributions

M.T.B. and S.K.D. conceptualization; M.T.B. experimental design and sample processing, data analysis, and manuscript writing; R.E.L. bioinformatics and data analysis; A.Q. and O.S. analytical sample runs; J.R.C. standards synthesis, M.H. standards synthesis and pathway analysis supervision; I.F. LOX analysis and pathway analysis supervision; I.F., C.A.O.,

A.R.G., D.J.S., V.M.W. manuscript editing; C.E.W. and S.K.D. funding, supervision, and manuscript editing.

## Funding

## Competing interests
The authors declare no competing interests.
