## [Transparent Peer Review file · Communications Biology]

Octadecanoids as emerging lipid mediators in cnidarian-dinoflagellate symbiosis

Corresponding Author: Dr Craig Wheelock

Version 0:

Reviewer comments:

Reviewer #1

(Remarks to the Author)

The manuscript from Botana et al describes the investigation of octadecanoids as signaling molecules involved in cnidarian-dinoflagellate symbiosis. The authors used chiral supercritical fluid chromatography coupled with MS/MS and deeply investigated the distinct stereospecificity in oxylipin synthesis between host and symbiont. In addition, the authors identify novel lipoxygenases in Symbiodiniaceae.

Overall, the manuscript is well written, and the literature is well cited and appropriate for the topics described in the manuscript. I have recommendations for clarifying some points in the manuscript, mainly related to the methodology and data quality assurance, which are described below.

Detailed comments:

- Supplementary Tables: Part of the tables are in the excel sheet, while another part is in the supplementary material PDF. This makes it confusing to find these pieces of information, so I suggest putting them all in Excel and leave the supplementary PDF for supplementary the figures only.

- Methods, line 397-398: "...1.5 mL of methanol (MeOH) and 10 µL of internal standard (IS) mix" which internal standard, and at what concentration?

- Methods, line 413-415: It is not clear if the FSW was also passed through the whole process of SPE and subsequently filtered as well.

- Methods, Statistical analysis: 1) Did the authors apply any type of data normalization/transformation for the multivariate analyses? 2) Which method was used for FDR correction?

- Methods, enantiomeric excess calculation: the ee calculation is well explained. However, how was the impact of the instrumental variability in the chiral SFC-MS/MS evaluated? Were technical replicates acquired for RSD calculation?

- Figure 3: it seems that the color light blue is different for A vs B/C. The legend of the colors can be moved to the bottom of all the plots and organized horizontally to be clearer that it refers to all the plots, and not just B.

- Figure 5: For all boxplots, show all the data points and boxplots whiskers.

Reviewer #2

(Remarks to the Author)

1. Brief summary of the manuscript

In the manuscript, the authors test the hypothesis that oxylipin signaling is responsible for inter-partner recognition and homeostasis regulation between cnidarian host and dinoflagellate symbionts. 84 octadecanoids were reported for sea anemone *Exaiptasia diaphana* and symbionts: *Breviolum minutum* and *Durusdinum trenchii* which are native and non-

native symbionts, respectively. Based on the differences in the stereochemistry of oxylipins authors propose that symbiosis with native symbiont is beneficial (less pro-inflammatory 9-HODE) and non-native symbiont causes more cellular stressed (higher level of autooxidation-derived products). In addition, sequences of putative symbiont LOXs were discovered from RNA seq datasets.

2. Overall impression of the work. If you recommend publication, please outline briefly what you consider to be the outstanding features.

It is a timely and very important study providing results for a large audience. For coral, symbiosis with zooxanthellae is an essential source of energy. Revealing distinct aspects of cell to cell signaling in the symbiosis process is crucial for advancing the field. Oxylipins are signaling molecules in the plants as well as in animals. Profiling which oxylipins are formed in which organism with proper controls is essential to understand and delineate the pathways present and how the lipid mediators contribute to symbiosis process. This manuscript is of interest to workers in the field and makes an impactful contribution to the literature.

3. Specific comments, with recommendations for addressing each comment

1) What is the PUFA content of sea anemone *Exaiptasia diaphana*? It would help the reader to estimate could the PUFAs of symbiont be readily converted by anemone dioxygenases, and whether there is a competition between host/symbiont PUFA pool.

2) What kind of lipoxygenases are present in *E. diaphana*? Lipoxygenases are substrate specific. Would *E. diaphana* lipoxygenase(s) use symbiont's PUFAs as substrates? Coral 11R-LOX substrate specificity is published (Mortimer 2006, <https://doi.org/10.1016/j.abb.2005.10.023>). Based on our unpublished data, coral 8-LOXs use AA, less EPA, dihomogamma-LA and adrenic acid and do not use linoleic acid; α -linolenic acid (C18 n-6); 11, 14, 17- eicosatrienoic acid (C18 n-3); 13, 16, 19-docosatrienoic acid (C22 n-3). We have not tested stearidonic acid (18:4 n-3). In theory, SDA could be converted by this coral enzyme but most likely less than the main substrate, AA.

3) Current experiments based on R to S- shift do give an indication that there could be exchange (AND most likely there is) but detecting the oxylipins doesn't provide direct proof of translocation of oxylipin from host to symbiont and vice versa.

4) For consistency, authors could use either *Aiptasia* or host to refer to the sea anemone *Exaiptasia diaphana* throughout the manuscript.

5) Sea anemones were fed during the experiment. How natural conditions (lack of food and relying mainly on light and symbiont) would affect the symbiosis with *B. minutum* and *D. trenchii* and the octadecanoid profile of the host/symbionts?

6) How opportunistic *E. diaphana* relates to coral reef building? Are you working towards other coral symbiosis models? If recolonization of coral reefs is the long-term aim of these studies, maybe using other coral models would be more suitable.

7) In future studies it would be of great interest to clone, and show the catalytic activity, regional specificity, and substrate preference of symbiodinium LOXs (the same is true for *E. diaphana* LOXs).

8) Figs. 1 and 2. were great addition to explain the concept and show the chemical structures for a common reader.

9) Fig. 3a. The Y-axis (ng/g dry biomass) should be adjusted to 6×10^3 as on Figs 3b and 3c to indicate, that there are less monohydroxy octadecanoids formed, more "exoxy" and "oxo" in *E. diaphana*.

10) Fig. S2. For clarity only one alignment should be presented. Please add Bm sequence to the second sequence alignment.

I would not dare to claim that the lipoxygenases are catalytically active based on the sequence data only. Please refer to them as putative 9S- and 13S-LOXs.

11) Table S7. Retention time of 18-HOTE is ".64". Correct?

12) Line 126 Add "of oxylipins" ...patterns of oxylipins in aposymbiotic...

13) Line 190. Add "putative" to the title as follows: Novel putative 9(S)- and 13(S)-lipoxygenases ...

14) Line 199. Rephrase: The alanine (Coffa determinant) and valine (Hornung sites) alignments with the active site residues of AtLOX 1 and 5 revealed ...

15) Lines 281-283. "In our experimental system, the abundance of the symbiont-derived 13(S)-HOTE decreased in the symbiont fraction and increased in the host fraction while in symbiosis, supporting evidence for inter-partner communication via translocation of this compound." Indeed, an increase-decrease in symbiont and in host was shown but not the actual translocation of oxylipins. Please rephrase accordingly.

16) Lines 308-311. Gene expression study based on previous RNA sequencing is described under methods, but results are

not presented. Please add (data not shown) to the sentence as follows: "The active constitutive expression of both 13(S)- and 9(S)-LOXs candidates across both cultured and symbiotic Symbiodiniaceae RNA-seq datasets (data not shown) also suggests... .

17) Lines 326-327. "provided insight into their biosynthetic route of formation" This has not been discussed, please rephrase e.g.,: "identified the host/symbiont octadecanoid patterns".

18) Line 327. "dynamic exchange". As no translocation experiments were conducted, replace "exchange" with "change".

Reviewer #3

(Remarks to the Author)

Oxylipin signaling is important component of immune regulation in mammals. In this work, authors have meticulously quantified different oxylipins in aposymbiotic anemone *Exaiptasia diaphana* (a model organism for cnidarian biology) and in *Exaiptasia* inoculated with either the native symbiont *Breviolum minutum* or the non-native symbiont *Durusdinium trenchii*. The work performed, observations made and discussion from the experimental results is well written and described while referencing known observations in corals, plants or humans as appropriate. The authors provide cogent arguments backed by literature and experiments to bolster their observations. The work is of high importance with the threat to coral symbiosis in the face of changing ocean temperatures and pollution. Few minor comments below.

1. There is only fungal sequence in the phylogenetic tree for LOX. Thus, the observation that symbiont-associated Lox are closest to fungi is not sound. The authors could add more fungal sequences to the tree to back their discussion.
2. Cell density was quantified by normalizing cell numbers by protein counts, which is also one of the methods used in mammalian cell culture. Since symbiont density measurements is crucial to the observations made in the study, there should be some mention of why this is the best method citing original literature where such methods were shown to be accurate.
3. In discussion, line 223: "to elucidate biosynthetic pathways in the cnidarian-dinoflagellate symbiosis": This is a bold statement since the pathways are proposed not elucidated. Without in enzyme activity reconstitution in vitro or in vivo, the biosynthetic pathway is not elucidated but proposed.

Version 1:

Reviewer comments:

Reviewer #1

(Remarks to the Author)

The authors made changes to improve the manuscript and their responses were appropriate. More experimental detail was given to ensure future reproducibility. However, the full data points were not included in the boxplots, as previously requested.

Reviewer #2

(Remarks to the Author)

All my questions, comments and suggestions were addressed, and I have no further questions-comments

Reviewer #3

(Remarks to the Author)

All my concerns are addressed appropriately.

Reviewers' comments:

Reviewer #1 (Remarks to the Author):

The manuscript from Botana et al describes the investigation of octadecanoids as signaling molecules involved in cnidarian-dinoflagellate symbiosis. The authors used chiral supercritical fluid chromatography coupled with MS/MS and deeply investigated the distinct stereospecificity in oxylipin synthesis between host and symbiont. In addition, the authors identify novel lipoxygenases in Symbiodiniaceae. Overall, the manuscript is well written, and the literature is well cited and appropriate for the topics described in the manuscript. I have recommendations for clarifying some points in the manuscript, mainly related to the methodology and data quality assurance, which are described below.

Detailed comments:

- Supplementary Tables: Part of the tables are in the excel sheet, while another part is in the supplementary material PDF. This makes it confusing to find these pieces of information, so I suggest putting them all in Excel and leave the supplementary PDF for supplementary the figures only.

This is a very good suggestion. We have now included all the **Supplementary Tables** in the Excel spreadsheet and all of the **Supplementary Figures** in the Supplemental Information PDF file.

- Methods, line 397-398: "...1.5 mL of methanol (MeOH) and 10 μ L of internal standard (IS) mix" \diamond which internal standard, and at what concentration?

This is a fair point. We had initially not included detailed method information because the method is already published (Quaranta, A. et al. *Analytical Chemistry*, (2022) 94(42), 14618). However, we agree that it is helpful to the reader to have increased analytical detail in this manuscript. We have therefore added a table showing the composition and concentration of the internal standard mix (**Table S1**). All the other **Supplementary Tables** have been re-ordered in the new version of the manuscript.

- Methods, line 413-415: It is not clear if the FSW was also passed through the whole process of SPE and subsequently filtered as well.

All blanks (*i.e.*, FSW only and cultured medium) were extracted and processed in the same way as for the experimental samples, which include SPE, filtering and reconstitution. We have rephrased this section in the manuscript to make it more clear. Please see **Lines 421-422**, to read:

"All blanks were extracted the same way as for of the experimental samples"

- Methods, Statistical analysis: 1) Did the authors apply any type of data normalization/transformation for the multivariate analyses? 2) Which method was used for FDR correction?

1) Yes, data preprocessing was performed. The octadecanoid dataset underwent log transformation for normalization prior to the principal component analysis (PCA) represented in **Figure 4**. The PCA relied on the Euclidean distance metric for capturing magnitude differences linked to absolute changes in the levels of the octadecanoids. To increase the clarity, additional details about data normalization and PCA analysis were added to the Statistical Analysis section of the **Methods**. Please see **Lines 512-513**, which now read:

“Before statistical analysis, the octadecanoid dataset was normalized using log transformation. Principal component analysis (PCA) using Euclidian distance metrics was used to identify the main differences in the octadecanoid profiles associated with symbiotic state and symbiont identity, in the host and symbiont separately”

2) We used the Benjamini-Hochberg (BH) method for adjustment of raw p-values and calculated FDR values using an R script component of the MetaboAnalyst software version 6.0. (Pang, AP., Luo, Y., He, C. et al (2020). A polyp-on-chip for coral long-term culture. *Sci Rep* 10, 6964). This information has now been added to the manuscript. Please see **Lines 540-542**, to read:

“Additionally, pairwise comparisons were analyzed with Tukey's HSD test, and the significance threshold set at $p < 0.05$ with false discovery rate corrections, using the Benjamini-Hochberg calculation method.”

- Methods, enantiomeric excess calculation: the ee calculation is well explained. However, how was the impact of the instrumental variability in the chiral SFC-MS/MS evaluated? Were technical replicates acquired for RSD calculation?
We are glad to know that the ee calculation was sufficiently explained because this was a concern we had in the drafting of the manuscript.

Technical replicates were not performed in our study because it is not a common practice in the lipidomics analytical community. This point was discussed in recently published guidelines (Schebb, N. H., et al. (2025). Technical recommendations for analyzing oxylipins by liquid chromatography–mass spectrometry. *Science signaling*, 18(887), eadw1245). Instead, the method employed for our study underwent rigorous validation that evaluated intra- and interday precision, intra- and interday accuracy, recovery, and matrix effects. Details are provided in the original method publication (Quaranta, A., et al. (2022). Development of a chiral supercritical fluid chromatography–tandem mass spectrometry and reversed-phase liquid chromatography–tandem mass spectrometry platform for the quantitative metabolic profiling of octadecanoid oxylipins. *Analytical Chemistry*, 94(42), 14618). A brief summary of the metrics reported in the paper is provided here: instrumental linearity was good ($R^2 > 0.995$), the lower limit of quantification (LLOQ) ranged from 0.03 to 6.00 ng/mL, the average accuracy ranged from 89 to 109%, the coefficients of variation (CV) were $< 14\%$ (at medium and high concentrations) and 26% (at low concentrations), the validation showed negligible matrix effects ($< 16\%$ for all analytes), and average recoveries ranged from 71 to 83%. Based upon these performance metrics, the octadecanoid SFC-MS/MS method was determined to be fit-for-purpose and as is common practice in the field, technical replicates were not acquired. However, it is important to note that of course biological replicates were acquired and the reported %RSD data in the paper are from biological replicates.

- Figure 3: it seems that the color light blue is different for A vs B/C. The legend of the colors can be moved to the bottom of all the plots and organized horizontally to be clearer that it refers to all the plots, and not just B.

We appreciate the Reviewer's attention to detail and have edited **Figure 3** accordingly.

- Figure 5: For all boxplots, show all the data points and boxplots whiskers.

The boxplot graphs were generated using GraphPad and the upper and lower limit of the plot constitute the interquartile space in addition to the maximum and minimum whisker, which results in the variable box-plot total length among the graphs in **Figure 5**. A better explanation was added to the figure's legend, to read:

“Box plot ranges represent the interquartile space in addition to the maximum and minimum whiskers with the median lane for each graph individually, explaining the differences in box plot total length among graphs”

We chose this plot for a cleaner representation of the data, given the complexity of information presented.

Reviewer #2 (Remarks to the Author):

1. Brief summary of the manuscript

In the manuscript, the authors test the hypothesis that oxylipin signaling is responsible for inter-partner recognition and homeostasis regulation between cnidarian host and dinoflagellate symbionts. 84 octadecaonids were reported for sea anemone *Exaiptasia diaphana* and symbionts: *Breviolum minutum* and *Durusdinum trenchii* which are native and non-native symbionts, respectively. Based on the differences in the stereochemistry of oxylipins authors propose that symbiosis with native symbiont is beneficial (less pro-inflammatory 9-HODE) and non-native symbiont causes more cellular stressed (higher level of autooxidation-derived products). In addition, sequences of putative symbiont LOXs were discovered from RNA seq datasets.

2. Overall impression of the work. If you recommend publication, please outline briefly what you consider to be the outstanding features. It is a timely and very important study providing results for a large audience. For coral, symbiosis with zooxanthellae is an essential source of energy. Revealing distinct aspects of cell to cell signaling in the symbiosis process is crucial for advancing the field. Oxylipins are signaling molecules in the plants as well as in animals. Profiling which oxylipins are formed in which organism with proper controls is essential to understand and delineate the pathways present and how the lipid mediators contribute to symbiosis process. This manuscript is of interest to workers in the field and makes an impactful contribution to the literature.

We are thankful to the Reviewer for supporting our manuscript. We also appreciate the constructive comments and suggestions. Please find below our responses to the specific comments.

3. Specific comments, with recommendations for addressing each comment

1) What is the PUFA content of sea anemone *Exaiptasia diaphana*? It would help the reader to estimate could the PUFAs of symbiont be readily converted by anemone dioxygenases, and whether there is a competition between host/symbiont PUFA pool. Thank you for highlighting this important consideration. To our knowledge, no studies have directly characterized the fatty acid or polyunsaturated fatty acid (PUFA) composition of *Exaiptasia diaphana* (Aiptasia). However, transcriptomic and proteomic studies (including those referenced in our manuscript, e.g., Oakley *et al.*, 2016; Matthews *et al.*, 2017; Gabay *et al.*, 2018) consistently identify lipid metabolism and transport as critical processes modulated during symbiosis establishment and under environmental stress. Our study focused on octadecanoid profiling, and the data (**Table S1**) revealed a striking disparity: HOTES, which derive from the PUFA

stearidonic acid (SDA, 18:4 n-3), are present at concentrations two orders of magnitude higher in symbionts compared to aposymbiotic *Aiptasia*. This strongly supports a symbiont origin for the precursor of the putatively bioactive 13(S)-HOTE discussed in our work. Importantly, this specific case shows no evidence of competition between host and symbiont PUFA pools.

We fully agree that broader PUFA profiling would provide valuable insight, particularly for understanding potential competition over shared substrates in other pathways. However, PUFA measurement was not possible using the analytical platforms in our study, which were optimized for oxylipin quantification (the primary focus of the study) and lacked the resolution to reliably monitor precursor fatty acids. Your comment underscores an important avenue and consideration for our future research. We now acknowledge this limitation and its importance for future studies in our **Discussion (Lines 344-346)**, which now reads:

“It would be valuable in future studies to also determine the PUFA content in both the symbiont and host, which could provide increased understanding of the biosynthesis pathways and potential inter-partner exchange and signaling of octadecanoids and other oxylipins.”

2) What kind of lipoxygenases are present in *E. diaphana*? Lipoxygenases are substrate specific. Would *E. diaphana* lipoxygenase(s) use symbiont’s PUFAs as substrates? Coral 11R-LOX substrate specificity is published (Mortimer 2006, <https://doi.org/10.1016/j.abb.2005.10.023>). Based on our unpublished data, coral 8-LOXs use AA, less EPA, dihomogamma-LA and adrenic acid and do not use linoleic acid; α -linolenic acid (C18 n-6); 11, 14, 17- eicosatrienoic acid (C18 n-3); 13, 16, 19- docosatrienoic acid (C22 n-3). We have not tested stearidonic acid (18:4 n-3). In theory, SDA could be converted by this coral enzyme but most likely less than the main substrate, AA.

We appreciate this insightful comment. To our knowledge, no studies have yet characterized lipoxygenase enzymes in *Exaiptasia diaphana* (*Aiptasia*). However, prior research (including our manuscript in **Lines 67-71**) highlights the translocation of fatty acids from dinoflagellate symbionts to the cnidarian host as a critical process for oxylipin synthesis and symbiosis homeostasis. This suggests that PUFAs of symbiont origin may serve as primary substrates for oxylipin production in *Aiptasia*. As also pointed out by the reviewer, AOS-LOX, 8(R)-LOX and 11(R)-LOX have been studied in other species of cnidarians reported to be involved in the synthesis of eicosanoids from C20 fatty acids, but with less binding for the oxidation of most C18 fatty acids, and SDA has never been tested. On the other hand, the novel putative symbiont LOX sequences showed a combination of amino acid residues at their active sites that strongly suggest them to be synthesis agents of some of the octadecanoids reported as having strong bioactivity in our study, such as 13(S)-HOTE. More clarification was added on **Lines 214-221**, to read:

“The alanine (Coffa determinant) alignment and valine (Hornung site) alignments with the active site residues of AtLOX 1 and 5 revealed that MSTRG.68745/7968 were 9(S)-LOXes (Fig. S2). All transcripts demonstrated active constitutive expression across all tested samples of both cultured and symbiotic Symbiodiniaceae RNA-seq datasets (Fig. S3). Homology comparisons grouped the new Symbiodiniaceae LOXes within their own clade in the tree of life, being evolutionarily divergent from green algal

(i.e., *Chlamydomonas reinhardtii* and *Lobosphaera incisa*) and vascular plant LOXes, but rather closely related to a fungal Mn-LOX sequences (Fig. 6).”

We have incorporated Mortimer et al. (2006) into our **References** and **Discussion** on acknowledging also the existence of 8(R) and 11(R) LOX on **Lines 258-261**, to read:

“In cnidarians, AOS and 8(R)- and 11(R) LOXes genes have been linked to eicosanoid synthesis^{40,58-60}; but whether they are also involved in the synthesis of TriHOMEs and other octadecanoids remains to be investigated, as does the potential synthesis role by EAS.”

While our study did not characterize LOX sequences in *Aiptasia*, this might be an interesting topic for our future research, given the importance of cross-kingdom enzyme-substrate interactions in mediating oxylipin signaling.

3) Current experiments based on R to S- shift do give an indication that there could be exchange (AND most likely there is) but detecting the oxylipins doesn't provide direct proof of translocation of oxylipin from host to symbiont and vice versa.

We fully agree with this Reviewer that we should be more careful in how we refer to these observations. Accordingly, whenever “translocation” or “exchange” of oxylipins was used in the text we reframed it as “putative” or “potential”. Please see **Lines 135, 194, 203, 239, 301, 303 and 353**.

4) For consistency, authors could use either *Aiptasia* or host to refer to the sea anemone *Exaiptasia diaphana* throughout the manuscript.

We thank the Reviewer for this comment. Alterations in the manuscript have been made accordingly, including in the figure legends. However, in some cases it is essential to explicitly refer to *Aiptasia* given that the reported interactions between symbionts and this host species cannot necessarily be generalised to other host cnidarians. For instance, *D. trenchii* is a non-native symbiont for *Aiptasia* but is found naturally in symbiosis with some other species. In other cases, though, “host” is a more appropriate term, where a host-symbiont interaction can be considered to be more general in nature. We have therefore been careful to be as consistent as possible while ensuring accuracy.

5) Sea anemones were fed during the experiment. How natural conditions (lack of food and relying mainly on light and symbiont) would affect the symbiosis with *B. minutum* and *D. trenchii* and the octadecanoid profile of the host/symbionts?

The sea anemone *Aiptasia* has only ever been observed in the symbiotic state in nature, and the contribution of heterotrophic feeding is essential for its optimal function and survival, both in the environment and in laboratory tanks. Indeed, if unfed in the symbiotic state, the anemone rapidly loses protein biomass, while the symbiotic algae lose chlorophyll, decline in population density and exhibit signs of nutrient stress. Given this, it is standard to feed these anemones during all but the most short-term experimental work (e.g., Hillyer, K. E.; Tumanov, S.; Villas-Bôas, S.; Davy, S. K. (2015) Metabolite Profiling of Symbiont and Host during Thermal Stress and Bleaching in a Model Cnidarian-Dinoflagellate Symbiosis. *Journal of Experimental Biology* and Matthews, J. L.; Crowder, C. M.; Oakley, C. A.; Lutz, A.; Roessner, U.; Meyer, E.; Grossman, A. R.; Weis, V. M.; Davy, S. K. (2017) Optimal Nutrient Exchange and Immune Responses Operate in Partner Specificity in the Cnidarian-Dinoflagellate

Symbiosis. PNAS, 114 (50), 13194–13199 (Reference 22), unless explicitly studying the impacts of starvation. The references above have now been added to the Experimental setup section, which now reads:

“All anemones were fed with Artemia sp. nauplii at the same time of day and the old water was changed with fresh FSW 8 h after feeding^{21,22}. Anemones were maintained under these conditions for 16 days prior to sampling and analyses”

6) How opportunistic *E. diaphana* relates to coral reef building? Are you working towards other coral symbiosis models? If recolonization of coral reefs is the long-term aim of these studies, maybe using other coral models would be more suitable.

This is a good question – and certainly goes to the heart of the study. The sea anemone *E. diaphana* (i.e., *Aiptasia*) is the primary model system for studying the cnidarian-dinoflagellate symbiosis. It was formally proposed as a model in 2008 by Weis et al (Weis, V. M., Davy, S. K., Hoegh-Guldberg, O., Rodriguez-Lanetty, M., & Pringle, J. R. (2008). Cell biology in model systems as the key to understanding corals. Trends in ecology & evolution, 23(7), 369-376) with the advantages of allowing easier manipulation and cultivation under laboratory conditions, including the fact that it can be rendered symbiont-free (aposymbiotic) in a lab setting, as in our study. *Aiptasia* indeed does not produce a calcium carbonate skeleton and is not a reef-building organism, therefore our findings are not directly translatable or comparable to the response of coral reefs, which are the organisms with the ultimate ecological relevance. Nevertheless, in all our previous proteomic and metabolomic analyses, for example in response to thermal stress, *Aiptasia* behaves almost identically to reef corals (e.g., Hillyer, K. E.; Tumanov, S.; Villas-Bôas, S.; Davy, S. K. (2015) Metabolite Profiling of Symbiont and Host during Thermal Stress and Bleaching in a Model Cnidarian-Dinoflagellate Symbiosis. Journal of Experimental Biology; Hillyer, K. E., Dias, D. A., Lutz, A., Wilkinson, S. P., Roessner, U., & Davy, S. K. (2017). Metabolite profiling of symbiont and host during thermal stress and bleaching in the coral *Acropora aspera*. Coral Reefs, 36(1), 105-118. We are also currently working on corals with respect to oxylipin signaling; however, this is out of the scope of this current manuscript.

A recent editorial by Roberty et al (Roberty, S., Weis, V. M., Davy, S. K., & Voolstra, C. R. (2024). Editorial: *Aiptasia*: a model system in coral symbiosis research. Front Mar Sci 11) further discussed the use of *Aiptasia* as a model system and the reference has now been added to our manuscript for better contextualization for the use of *Aiptasia* in **Lines 124-127**, to read:

*“We used the sea anemone *Exaiptasia diaphana* (commonly called ‘Aiptasia’) as a host model – a, which is widely used model system in cnidarian-dinoflagellate symbiosis studies³⁶ – when, in symbiosis with either its native (homologous) symbiont *Breviolum minutum* or the non-native (heterologous) symbiont *Durusdinium trenchii*.”*

7) In future studies it would be of great interest to clone, and show the catalytic activity, regional specificity, and substrate preference of symbiodinium LOXs (the same is true for *E. diaphana* LOXs).

We thank the reviewer for this remark and definitely agree that the given suggestion would be of high interest for our future research and for validating the novel putative enzyme sequences elucidated in the current study. We acknowledge this point in the

Discussion Lines 329-333, and also mention that such an approach and methodology was used in one of our previous publications (Djian, B.; Hornung, E.; Ischebeck, T.; Feussner, I. 2019 The Green Microalga *Lobosphaera Incisa* Harbours an Arachidonate 15 S -lipoxygenase. *Plant Biol J*, 21 (S1), 131–142) characterizing a new LOX in the commercially valuable microalgae *Lobosphaera incisa*.

“Further characterization of the enzyme sequences presented here for the exploring of catalytic activity, regional specificity and fatty acid substrate preference are still needed for validation and might be promising for future research using a similar approach as in one of our previous studies characterizing a new LOX in Lobosphaera incisa⁸².”

8) Figs. 1 and 2. were great addition to explain the concept and show the chemical structures for a common reader.

We are grateful to the Reviewer for the positive comment.

9) Fig. 3a. The Y-axis (ng/g dry biomass) should be adjusted to 6×10^3 as on Figs 3b and 3c to indicate, that there are less monohydroxy octadecanoids formed, more “expoxy” and “oxo” in *E. diaphana*.

This was a good point – thank you. All the Y-axes in the new version of **Figure 3** were adjusted to 6×10^3 for standardization, as suggested.

10) Fig. S2. For clarity only one alignment should be presented. Please add Bm sequence to the second sequence alignment. I would not dare to claim that the lipoxygenases are catalytically active based on the sequence data only. Please refer to them as putative 9S- and 13S-LOXs.

We appreciate the Reviewer’s comments and changed for only one sequence alignment as suggested (please see new version of **Fig. S2**). We agree that more investigation is needed to confirm the existence and the catalytical activity of the new putative enzyme sequences. We have re-phrased this to “putative enzymes” in the legend of **Figure S2** and in the manuscript text (e.g., see **Line 239**).

11) Table S7. Retention time of 18-HOTE is “.64”. Correct?

Thank you – that was a very good catch. The retention time was corrected to 4.64 min in the new version of **Table S7**.

12) Line 126 Add “of oxylipins” ...patterns of oxylipins in aposymbiotic...

Changed as requested.

13) Line 190. Add “putative” to the title as follows: Novel putative 9(S)- and 13(S)-lipoxygenases ...

Changed as requested.

14) Line 199. Rephrase: The alanine (Coffa determinant) and valine (Hornung sites) alignments with the active site residues of AtLOX 1 and 5 revealed ...

Changed as requested.

15) Lines 281-283. “In our experimental system, the abundance of the symbiont-derived 13(S)-HOTE decreased in the symbiont fraction and increased in the host fraction while in symbiosis, supporting evidence for inter-partner communication via

translocation of this compound.” Indeed, an increase-decrease in symbiont and in host was shown but not the actual translocation of oxylipins. Please rephrase accordingly. The statement was rephrased following the reviewer recommendation, to read:

“In our experimental system, the abundance of the symbiont-derived 13(S)-HOTE decreased in the symbiont fraction and increased in the host fraction while in symbiosis, supporting evidence for inter-partner communication via a putative translocation of this compound. While confirmation is still needed, our findings support this octadecanoid-receptor binding as a potential mechanism of host immune modulation”

16) Lines 308-311. Gene expression study based on previous RNA sequencing is described under methods, but results are not presented. Please add (data not shown) to the sentence as follows: “The active constitutive expression of both 13(S)- and 9(S)-LOXs candidates across both cultured and symbiotic Symbiodiniaceae RNA-seq datasets (data not shown) also suggests... .

We thank the reviewer for this remark. The data confirming active expression of the novel sequences have been added to the **Results** in **Lines 216-221** and **Figure S3**, to read:

*“All transcripts demonstrated active constitutive expression across all tested samples of both cultured and symbiotic Symbiodiniaceae RNA-seq datasets (Fig. S3). Homology comparisons grouped the new Symbiodiniaceae LOXes within their own clade in the tree of life, being evolutionarily divergent from green algal (i.e., *Chlamydomonas reinhardtii* and *Lobosphaera incisa*) and vascular plant LOXes, but rather closely related to a fungal Mn-LOX sequences (Fig. 6).”*

To determine the expression of LOX candidates, un-normalised counts of genome-aligned RNA reads mapped to LOX mRNA were summarised at the gene level for each sample using FeatureCounts (Liao et al. Bioinformatics, 30(7), 2014, 923–930) and tested for different expression between conditions performed with DeSeq2, as previously described. The new reference and details have been added to the **Methods** section (**Lines 510-514**), to read:

“All counted genes were tested for differential expression among experimental conditions from the original raw datasets using DeSeq2, with default internal gene-filtering procedures (see Table S10 for RNA-seq libraries used for LOX gene expression counts). Before expression testing, counted genes were pre-filtered to retain only genes with at least 10 un-normalized reads counted in every sample.”

17) Lines 326-327. “provided insight into their biosynthetic route of formation” This has not been discussed, please rephrase e.g.,: “identified the host/symbiont octadecanoid patterns”.

We have replaced the initial phrasing. Please see **Lines 349-351**, which now read:

“This work presents the first detailed quantification of octadecanoids in cnidarians and their associated symbionts and reveals the stereochemistry specificity of octadecanoid biosynthesis in each studied species.”

18) Line 327. “dynamic exchange”. As no translocation experiments were conducted, replace “exchange” with “change”.
Changed as requested.

Reviewer #3 (Remarks to the Author):

Oxylipin signaling is important component of immune regulation in mammals. In this work, authors have meticulously quantified different oxylipins in aposymbiotic anemone *Exaiptasia diaphana* (a model organism for cnidarian biology) and in *Exaiptasia* inoculated with either the native symbiont *Breviolum minutum* or the non-native symbiont *Durusdinium trenchii*. The work performed, observations made and discussion from the experimental results is well written and described while referencing known observations in corals, plants or humans as appropriate. The authors provide cogent arguments backed by literature and experiments to bolster their observations. The work is of high importance with the threat to coral symbiosis in the face of changing ocean temperatures and pollution. Few minor comments below.

1. There is only fungal sequence in the phylogenetic tree for LOX. Thus, the observation that symbiont-associated Lox are closest to fungi is not sound. The authors could add more fungal sequences to the tree to back their discussion.

This is an excellent point. We have added three new fungal sequences to the phylogenetic tree (total of 4). Please see the new version of this tree in **Figure 6**. The new sequences were all clustered close to the symbiont’s putative LOX sequences, in further support of our observation about the evolutionary proximity of the symbiont and the fungal Mn-LOX enzymes. The new sequences were obtained from NCBI from the species *Fusarium oxysporum*, *Aspergillus fumigatus* and *Nakataea oryzae*. Access number details were added to **Table S9**.

2. Cell density was quantified by normalizing cell numbers by protein counts, which is also one of the methods used in mammalian cell culture. Since symbiont density measurements is crucial to the observations made in the study, there should be some mention of why this is the best method citing original literature where such methods were shown to be accurate.

Estimating symbiont population via normalizing symbiont cell counts by host protein content is the standard protocol in the cnidarian-dinoflagellate symbiosis and coral biology fields. While some other protocols do exist, such as normalising symbiont numbers by coral surface area or estimating symbiont chlorophyll autofluorescence in defined regions of confocal microscopy images, they are not as accurate as protein normalization when considering the whole symbiosis. There are no papers directly comparing these methods; however, to reassure general readers that this is a valid, reliable approach, we have cited a couple of recent papers where this same method has been used to measure the densities of different symbiont species in *Aiptasia* (Gorman et al. (2025) (Gorman, L. M., Tivey, T. R., Raymond, E. H., Ashley, I. A., Oakley, C. A., Grossman, A. R., ... & Davy, S. K. (2025). Stability of the cnidarian–dinoflagellate symbiosis is primarily determined by symbiont cell-cycle arrest. *Proceedings of the National Academy of Sciences*, 122(14), e2412396122) and Lust et al (2025) (Lust, B., Matthews, J. L., Oakley, C. A., Lewis, R. E., Mendis, H., Peng, L., ... & Davy, S. K. (2025). The Influence of Symbiont Identity on the Proteomic and Metabolomic Responses of the Model Cnidarian *Aiptasia* to Thermal Stress. *Environmental Microbiology*, 27(3), e70073).

3. In discussion, line 223: “to elucidate biosynthetic pathways in the cnidarian-dinoflagellate symbiosis”: This is a bold statement since the pathways are proposed not elucidated. Without in enzyme activity reconstitution in vitro or in vivo, the biosynthetic pathway is not elucidated but proposed.

We appreciate this observation and agree that no biosynthetic pathway was elucidated in our study and that enzyme activity reconstitution would be an interesting and essential task for this confirmation in future studies. We have rephrased the statement accordingly in **Lines 237-240**, to read:

“To our knowledge, this is the first study of LOX enzymes in Symbiodiniaceae and applied top-down bioinformatics to obtain novel enzyme sequences putatively associated with octadecanoid biosynthesis in the cnidarian-dinoflagellate symbiosis.”